

# miR-34a-5p inhibits the malignant progression of KSHV-infected SH-SY5Y cells by targeting c-fos

Shuyuan Wu[1,*], Zhaofu Wu[1,*], Huiling Xu[1], Jinli Zhang[1], Wenyi Gu[2], Xiaohua Tan[3], Zemin Pan[1], Dongdong Cao[1], Dongmei Li[1], Lei Yang[3], Dongmei Li[1] and Yuanming Pan[4]

[1] Key Laboratory of Xinjiang Endemic and Ethnic Diseases/NHC Key Laboratory of Prevention and Treatment of Central Asia High Incidence Diseases, School of Medicine, Shihezi University, Shihezi, Xinjiang, China
[2] Australian Institute for Bioengineering and Nanotechnology (AIBN), University of Queensland (UQ), St Lucia, Brisbane, Australia
[3] School of Medicine, Hangzhou Normal University, Hangzhou, Zhejiang, China
[4] Department of Cellular and Molecular Biology, Beijing Chest Hospital, Capital Medical University, Beijing Tuberculosis and Thoracic Tumor Research Institute, Beijing, Beijing, China
* These authors contributed equally to this work.

Corresponding authors
Dongmei Li, lidong_abc@126.com
Yuanming Pan,
peter.f.pan@hsc.pku.edu.cn

## ABSTRACT

**Background:** We aimed to investigate the effects of miR-34a-5p on c-fos regulation mediating the malignant behaviors of SH-SY5Y cells infected with Kaposi's sarcoma-associated herpesvirus (KSHV).

**Methods:** The KSHV-infected (SK-RG) and uninfected SH-SY5Y parent cells were compared for differentially expressed miRNAs using transcriptome sequencing. Then miR-34a-5p was upregulated in SK-RG cells by the miRNA mimics transfection. Cell proliferation ability was determined by MTT and plate clone assays. The cell cycle was assessed by flow cytometry analysis, and CDK4, CDK6, cyclin D1 levels were determined by Western blot analysis. The migration behavior was detected by wound healing and transwell assays. The protein levels of MMP2 and MMP9 were measured by Western blot analysis. The regulation of c-fos by miR-34a-5p was detected by the dual-luciferase reporter gene assay. Rescue assays were carried out by upregulating c-fos in miR-34a-5p-overexpressing SK-RG cells. KSHV DNA copy numbers and relative virus gene expressions were detected. Xenograft tumor experiments and immunohistochemistry assays were further used to detect the effects of miR-34a-5p.

**Results:** miR-34a-5p was lower in SK-RG cells. Restoration of miR-34a-5p decreased cell proliferation and migration, leading to a G1 cell cycle arrest and down-regulation of CDK4/6, cyclin D1, MMP2, MMP9. KSHV copy number and expression of virus gene including latency-associated nuclear antigen (LANA), replication and transcription activator (RTA), open reading frame (K8.1), and KSHV G protein-coupled receptor (v-GPCR) were also reduced. Furthermore, c-fos is the target of miR-34a-5p, while enhanced c-fos weakened cellular behaviors of miR-34a-5p-overexpressing cells. Xenograft experiments and immunohistochemistry assays showed that miR-34a-5p inhibited tumor growth and virus gene expression.
**Conclusion:** Upregulated miR-34a-5p in KSHV-infected SH-SY5Y cells suppressed cell proliferation and migration through down-regulating c-fos. miR-34a-5p was a candidate molecular drug for KSHV-infected neuronal cells.

# INTRODUCTION

Kaposi's sarcoma-associated herpesvirus (KSHV), also known as human herpesvirus 8 (HHV-8), is the etiologic agent of Kaposi's sarcoma, primary effusion lymphoma, and multicentric Castleman disease (*Ueda, 2018*).

Research has made great progress on the pathogenicity and molecular mechanisms of KSHV. The cycle of KSHV infection consists of two stages: transient lytic replication and persistent latent infection (*Broussard & Damania, 2020*). Many host cells lysed and died during lytic replication, while latent infection caused host cell malignant transformation and tumor formation. These two modes of infection have feedback regulation and can be transformed into each other (*Aneja & Yuan, 2017*; *Manners et al., 2018*). However, the virus exists mainly in a latent state to ensure its survival. *In vitro*, the infected cells require phorbol ester or sodium butyrate to induce the lytic state (*Chang et al., 2017*).

Patients with acquired immunodeficiency syndrome (AIDS) are susceptible to KSHV, and about 50% of patients develop neurological symptoms (*Cesarman et al., 2019*; *Singer & Thames, 2016*). The herpes viruses have shown a neurotropic tendency in recent years. Therefore, problems related to KSHV infection on nerve cells raise a concern. Previous studies on brain tissues from KSHV infected patients confirmed that KSHV existed in the brain parenchyma and mainly infected neurons in the clinic (*Tso et al., 2017*). We used KSHV to infect SH-SY5Y neuronal cells in a previous study. Infected cells were found to undergo spontaneous periodic lysis without induction, and cells could survive and be cultured (*Kong et al., 2021*). Hence, neuronal cells infected by KSHV need further exploration.

MicroRNAs (miRNAs) are single-stranded noncoding RNAs containing 20–25 nucleotides that regulate gene expression at the post-transcriptional level by binding to the 3′-untranslated regions (UTR) of mRNAs (*Kontomanolis et al., 2016*; *Saliminejad et al., 2019*). We performed transcriptome sequencing on KSHV-infected and uninfected SH-SY5Y cells to find differentially expressed miRNAs in KSHV-infected neuronal cells and found that miR-34a-5p was down-regulated in KSHV-infected SH-SY5Y cells. miR-34a-5p was predicted to regulate c-fos expression negatively. Based on previous studies, the import transcription factor c-fos was upregulated in KSHV-infected neuronal cells and promoted KSHV lytic replication and viral gene expression (*Xu, Cao & Li, 2020*; *Jaworski, Kalita & Knapska, 2018*; *Kovács, 1998*), miR-34a-5p has the additional research value.

We upregulated the level of miR-34a-5p and investigated the effects on the proliferation and migration of KSHV-infected SH-SY5Y cells. To explore the target regulation of miR-34a-5p *via* c-fos, we carried out the rescue assay and xenograft tumor experiments to identify that miR-34a-5p was a candidate molecular drug for inhibiting the malignancy of KSHV-infected neuronal cells. These findings could offer some novel insights into anti-KSHV research.

## MATERIALS AND METHODS

### Cell culture

SH-SY5Y and 293T cell lines were purchased from Shanghai Cell Bank, Chinese Academy of Sciences (Beijing, China). SH-SY5Y and 293T cells were cultured in Dulbecco's Modified Eagle Medium (DMEM; Gibco, Life Technologies, Carlsbad, CA, USA) containing 10% heat-inactivated fetal bovine serum (FBS; Hyclone, GE Healthcare, Little Chalfont, UK), with 1% 100× penicillin-streptomycin solution (Biosharp, Beijing, China). SH-SY5Y infection by recombinant KSHV (rKSHV.219) was performed previously (*Kong et al., 2021*). The KSHV-infected SH-SY5Y cells were named SK-RG. SK-RG cells were grown in DMEM supplemented with 10% heat-inactivated FBS, 1% 100× penicillin-streptomycin solution, and 6 μg/ml puromycin (Sigma-Aldrich, St. Louis, MO, USA), in an incubator at 37 °C and 5% $CO_2$.

### Screening the differentially expressed miRNAs

RNA was extracted from KSHV-infected (SK-RG) or uninfected SH-SY5Y cells. Transcriptome sequencing was performed in BGI Biotech, Shenzhen, China. Each group had three parallel samples. After filtering the original sequencing data, high-quality data were compared. The data were integrated to identify and analyze the miRNAs found to be differentially expressed, and the meaningful upregulated and downregulated miRNAs were shown in Table 1. In a data set of miRNAs downregulated in KSHV-infected neurons, we predicted some miRNAs potentially associated with mRNA profilings using TargetScan/miRanda/RNAhybrid websites for bioinformatics analyses. Altogether, the fold change value was taken to select the differential miRNAs with high statistical significance. miR-34a-5p was selected for further studies. Moreover, we performed further experiments to determine the exact interactions because miR-34a-5p might regulate c-fos expression.

### RNA extraction and real-time PCR

SH-SY5Y and SK-RG cells were seeded in 60-mm culture dishes and grown to 70–90% confluence for further study. Total RNA was extracted using TRIzol reagent (Invitrogen, Camarillo, CA, USA). RNA concentration and purity were detected. A RevertAid First Strand cDNA Synthesis kit (Thermo Fisher, Carlsbad, CA, USA) was utilized for cDNA synthesis. Real-time PCR was used to detect cDNA recovered (SYBR Green PCR Kit; Qiagen, Hilden, Germany). The following conditions were used to detect mRNA expression: 95 °C, 300 s; 95 °C, 10 s; 55 °C, the 30 s; and 72 °C, 35 s; for a total of 35 cycles.

**Table 1 The differentially expressed miRNA.**

| miRNAs | Expression | log2Ratio | Value |
|---|---|---|---|
| hsa-miR-30a-5p | Down | −1.03 | $P < 0.01$ |
| hsa-miR-24-3p | Down | −1.06 | $P < 0.01$ |
| hsa-miR-34a-5p | Down | −1.09 | $P < 0.01$ |
| hsa-miR-378g | Down | −1.41 | $P < 0.01$ |
| hsa-miR-140-3p | Down | −2.27 | $P < 0.01$ |
| hsa-miR-135b-5p | Down | −1.75 | $P < 0.01$ |
| hsa-miR-29a-3p | Down | −1.30 | $P < 0.01$ |
| hsa-let-7f-5p | Down | −1.45 | $P < 0.01$ |
| hsa-miR-28-3p | Down | −1.45 | $P < 0.01$ |
| hsa-miR-200a-3p | Down | −2.78 | $P < 0.01$ |
| hsa-miR-98-5p | Down | −2.04 | $P < 0.01$ |
| hsa-miR-210-3p | Down | −2.05 | $P < 0.01$ |
| hsa-miR-449a | Down | −2.21 | $P < 0.01$ |
| hsa-miR-486-5p | Down | −1.20 | $P < 0.01$ |
| hsa-miR-193a-3p | Down | −2.94 | $P < 0.01$ |
| hsa-miR-296-5p | Up | 8.82 | $P < 0.01$ |
| hsa-miR-130b-5p | Up | 4.49 | $P < 0.01$ |
| hsa-miR-375-3p | Up | 8.61 | $P < 0.01$ |
| hsa-miR-491-5p | Up | 4.46 | $P < 0.01$ |
| hsa-miR-128-3p | Up | 3.78 | $P < 0.01$ |
| hsa-miR-19a-3p | Up | 5.76 | $P < 0.01$ |
| hsa-miR-18a-3p | Up | 6.12 | $P < 0.01$ |
| hsa-miR-224-5p | Up | 6.26 | $P < 0.01$ |
| hsa-miR-18a-5p | Up | 5.83 | $P < 0.01$ |
| hsa-miR-503-5p | Up | 3.20 | $P < 0.01$ |
| hsa-miR-7-5p | Up | 3.10 | $P < 0.01$ |
| hsa-miR-339-3p | Up | 3.96 | $P < 0.01$ |
| hsa-miR-582-5p | Up | 4.73 | $P < 0.01$ |
| hsa-miR-92a-3p | Up | 5.27 | $P < 0.01$ |
| hsa-miR-218-5p | Up | 3.13 | $P < 0.01$ |
| hsa-miR-92b-3p | Up | 8.05 | $P < 0.01$ |
| hsa-miR-652-5p | Up | 6.10 | $P < 0.01$ |
| hsa-miR-32-3p | Up | 3.65 | $P < 0.01$ |
| hsa-miR-17-3p | Up | 5.47 | $P < 0.01$ |
| hsa-miR-133a-3p | Up | 5.58 | $P < 0.01$ |
| hsa-miR-200c-3p | Up | 4.11 | $P < 0.01$ |
| hsa-miR-126-5p | Up | 7.78 | $P < 0.01$ |
| hsa-miR-330-3p | Up | 6.64 | $P < 0.01$ |
| hsa-miR-193b-3p | Up | 9.08 | $P < 0.01$ |
| hsa-miR-20a-5p | Up | 5.28 | $P < 0.01$ |
| hsa-miR-1-3p | Up | 7.26 | $P < 0.01$ |

| Table 1 (continued) | | | |
|---|---|---|---|
| miRNAs | Expression | log2Ratio | Value |
| hsa-miR-505-3p | Up | 9.23 | $P < 0.01$ |
| hsa-miR-338-5p | Up | 5.54 | $P < 0.01$ |
| hsa-miR-153-3p | Up | 9.33 | $P < 0.01$ |
| hsa-miR-1304-3p | Up | 9.60 | $P < 0.01$ |

**Table 2 The sequences of primers.**

| Target gene | Primer sequence |
|---|---|
| hsa-miR-34a-5p RT | GTCGTATCCAGTGCAGGGTCCGAG |
| | GTATTCGCACTGGATACGACACAACC |
| hsa-miR-34a-5p-F | CGCGTGGCAGTGTCTTAGCT |
| hsa-miR-34a-5p-R | AGTGCAGGGTCCGAGGTATT |
| c-fos-F | AATCCGAAGGGAAAGGAATAAGA |
| c-fos-R | GTCTCCGCTTGGAGTGTATCAGT |
| v-GPCR-F | GTGCCTTACACGTGGAACGTT |
| v-GPCR-R | GGTGACCAATCCATTTCCAAGA |
| K8.1-F | AAAGCGTCCAGGCCACCACAGA |
| K8.1-R | GGCAGAAAATGGCACACGGTTAC |
| RTA-F | GAGTCCGGCACACTGTACC |
| RTA-R | AAACTGCCTGGGAAGTTAACG |
| LANA-F | AGCCACCGGTAAAGTAGGAC |
| LANA-R | GATGTGACCTTGGCGATGAC |
| β-actin-F | CGGAACCGCTCATTGCC |
| β-actin-R | ACCCACATCGTGCCCATCTA |
| U6 RT | GTCGTATCCAGTGCAGGGTCCGAG |
| | GTATTCGCACTGGATACGACAAAATA |
| U6-F | AGAGAAGATTAGCATGGCCCCTG |
| U6-R | ATCCAGTGCAGGGTCCGAGG |
| ORF 26 for Taq-man-F | CGAATCCAACGGATTTGACCTC |
| ORF26 for Taq-man-R | CCCATAAATGACACATTGGTGGTA |
| ORF 26 probe | 5′FAM/CCCATGGTCGTGCCGCAGCA/3′BHQ-1 |

We used the comparative $2^{-\Delta\Delta Ct}$ way to analyze the relative mRNA expression. Table 2 contained the primer sequences.

## Cell transfection

Lipofectamine 2000 was used to transfect SK-RG cells at the logarithmic growth stage. (Invitrogen, Camarillo, CA, USA). miRNA negative control (miR-NC) and miR-34a-5p mimics (miR-34a-5p) were generated by GenePharma (Shanghai, China). pcDNA3.1

vector (pcDNA3.1) and the c-fos eukaryotic expression vector (c-fos) were generated by Gene Create (Wuhan, China), which were diluted and transfected into SK-RG cells.

## Cell viability assay

Cell proliferation was measured using 3-(4,5-dimethylthiazol-2-yl)-2,5-diphenyltetrazolium bromide (MTT) assay (Solarbio, Beijing, China). In 96-well plates, $1 \times 10^3$ transfected SK-RG cells were seeded with a 100 µL suspension in each well. Cell proliferation was measured at different time points (24, 48, 72, 96, and 120 h). After incubating for 4 h with MTT, 150 µL dimethyl sulfoxide (DMSO, Solarbio, Beijing, China) was added and shaken for 15 min. A microplate reader was used to measure the absorbance at 490 nm.

## Plate clone assay

Cell proliferation was assessed by plate clone assay. Transfected SK-RG cells were digested and seeded into six-well plates (310 cells/well) in 1.5 ml of complete medium. The cells were incubated for 13 days, with the solution being changed every 3–4 days throughout the process. After discarding the culture media, it was washed twice with phosphate-buffered saline (PBS) and fixed with 4% paraformaldehyde for 40 min, then stained with 0.1% crystal violet and washed twice with PBS after 30 min. The culture plates were dried at room temperature and observed under an inverted microscope to count the number of cell clones formed in each group.

## Wound healing assay

Cells were spread into cell culture dishes and transfected. After culturing for 24 h, the density of cells was about 90%. We used a sterile pipette tip (1 ml) to scratch the monolayer. The cells were washed twice with PBS and incubated in 2% serum DMEM for further culture. Plates were photographed using a microscope 0, 12, 36, and 48 h after scratching at an identical location.

## Transwell migration assay

Cell migration was assessed by transwell assay. After 24 h of transfection, the cells were digested. Then, 1 ml of PBS and 1 ml of DMEM were added sequentially for washing. The cells were resuspended in DMEM and counted until a $2 \times 10^5$ cells/ml density was reached. Moreover, 200 µL of cell suspension was added to the upper chamber, and 600 µl of the medium with 10% FBS was added to the basolateral chamber. The chambers were incubated for 36 h at 37 °C. The washed cells were fixed with paraformaldehyde for 45 min and stained with 0.1% crystal violet for 35 min. Penetrating cells in three randomly selected fields of each sample were captured for counting under a light microscope. The experiment was repeated three times.

## Dual-luciferase reporter assay

The binding sites of miR-34a-5p and c-fos 3′-UTR were predicted using StarBase 2.0 (http://starbase.sysu.edu.cn/). Luciferase reporter constructs encoding the wild-type 3′-UTR of c-fos (pmirGLO-c-fos-wt) or mutant 3′-UTR of c-fos (pmirGLO-c-fos-mut)

were synthesized by GenePharma (Shanghai, China). Then, pmirGLO-c-fos-wt, pmirGLO-c-fos-mut, and 293T cells were co-transfected with miR-34a-5p mimics or miR-NC. After 24 h of incubation, the cells were collected and lysed. Using Renilla luciferase activity as an internal reference, dual-luciferase reporter gene analysis was used to calculate the relative activity of luciferase.

## Western blot analysis

After transfection for 48 h, SK-RG cells were collected and lysed for total protein extraction. The proteins were electrophoresed in sodium dodecyl sulfate-polyacrylamide gels and subsequently transferred to polyvinylidene difluoride membranes. The membrane was blocked with 5% skimmed milk for 2 h, and primary antibodies, including β-actin (1:1,000; ZSGB-Bio, Beijing, China), c-fos (1:1,000; Bioss, Beijing, China), CDK4, CDK6 (1:1,000; Bioworld Technology, St Louis Park, MN, USA), cyclin D1 (1:500; Bioworlde Technology, St Louis Park, MN, USA), matrix metalloproteinase (MMP)2 (1:1,000; Bioss, Beijing, China), and MMP9 (1:1,000; Boster, Wuhan, China), were added overnight at 4 °C. The membranes were washed with 1 × Tris-buffered saline with Tween solution. Following that, secondary goat anti-rabbit or goat anti-mouse antibodies conjugated with horseradish peroxidase (1:10,000; ZSGB-Bio, Beijing, China) were added for 2 h at room temperature. After washing the membranes three times with 1 × PBS, the bands were detected by electrochemiluminescence and analyzed using ImageJ software (1.48v; NIH, Bethesda, MD, USA).

## Flow cytometry analysis

Cell cycle distribution was detected using flow cytometry (FCM) assay. After transfection for 24 h, SK-RG cells were digested with trypsin, centrifuged at 1,000 r/min for 5 min, and then the cells were collected and fixed in 4 ml cooled 70% anhydrous ethanol. After being stored at −20 °C for 2 days, the fixed cells were centrifuged for 5 min and then hydrated with 3 ml PBS at room temperature for 15 min. After adding 1 ml of DNA staining solution, cells were incubated at 37 °C in the dark for 30 min, and the cell cycle was examined by FCM (BD Bio-sciences, San Jose, CA, USA).

## Taq-man real-time PCR

Taq-man real-time PCR was used to determine virus DNA copy number. The experiment was performed after transfection in SK-RG cells for 48 h, and each group DNA was extracted according to the instructions of the TIANGEN Cell Genome Kit. DNA concentration was measured and detected using a Taq-man real-time PCR kit (Takara Biomedical Technology, Beijing, China). The sensitivity was determined by testing decreasing DNA quantities of the ORF26 plasmid (10-fold dilutions from $10^3$ to $10^7$ ng/μl). The temperature was set to 95 °C 20 s; 95 °C 3 s, 60 °C 30 s, a total of 40 cycles. Primer sequences are shown in Table 2.

## Nude mice xenograft tumor assay

The xenograft tumor assay and ethics were approved by the institutional guidelines of the Animal Care and Use Committee at Laboratory Animal Research Center, Shihezi University (Xinjiang, China). BALB/c female nude mice were aged 4–6 weeks (mature

body organs and good vital health index) were purchased and fed in the Experimental Animal Research Center of Xinjiang Medical University. The individual mouse was considered the experimental unit within the studies. The nude mice were housed in a dedicated SPF facility and raised in a laminar air chamber with the food and drinking water sterilized. We estimated the degree of freedom (E) of an OVA and initially selected 14 mice for the experiment. The xenograft mouse tumor model was established by injecting $3 \times 10^6$ SK-RG cells into one side of the spine of nude mice. Before the intervention, six mice did not develop tumors and were euthanized. Then, tumor-bearing mice were randomized into two groups by computer generation of random numbers. The two groups were isolated and cultured in two culture rooms in the same environment to reduce confounding factors. In addition, the experimenters were blinded to the grouping. The mice in the experimental group ($n = 4$) were injected with miR-34a-5p agomir, while the control group ($n = 4$) were injected with miR-NC agomir. Injections were given every other day for 2 weeks. All mice were monitored closely. The tumor sizes and body weights of nude mice were measured before injection. Tumor growth was monitored continuously, and all mice were sacrificed by cervical dislocation after injection for two weeks, which was properly performed by trained professionals to avoid pain to subjects. After sacrificing the mice, tumors were collected and photographed.

## Immunohistochemical analysis

Tissues isolated from different groups of nude mice were fixed with 10% formalin and paraffin-embedded to produce tissue sections. After dewaxing the slices, they were boiled for 8 min in sodium citrate buffer (pH = 6.0). The sections were cooled to 30 °C and incubated in 3% hydrogen peroxide at room temperature for 10 min while being protected from light. The sections were washed with PBS ($3 \times 5$ min). Primary antibodies c-fos (1:100; Bioss, Beijing, China), CDK6, and cyclin D1 (1:50; Bioworld Technology, St Louis Park, MN, USA) were applied overnight at 4 °C. The sections were washed with PBS. The secondary antibody (ZSGB-Bio, Beijing, China) was added and incubated for 30 min at 37 °C. After washing again, the slides were colored with 3,3′-diaminobenzidine (DAB) working solution for 60 s at room temperature, and finally, the slides were re-stained with hematoxylin and sealed.

## Statistical analyses

Experimental results were repeated at least three times in every group. The differences between the two groups were compared by the Student $t$-test. GraphPad Prism 5.0 was used to conduct all statistical analyses (GraphPad Software, San Diego, CA, USA). A statistically significant difference was marked by a $P$-value < 0.05.

# RESULTS

## The miR-34a-5p level was reduced in SK-RG cells

We infected the SH-SY5Y cells as described previously, and the infected cells were named SK-RG (Fig. 1A). The differentially expressed miRNAs were screened in KSHV-infected and uninfected SH-SY5Y cells by RNA sequencing, of which 1,462 miRNAs were

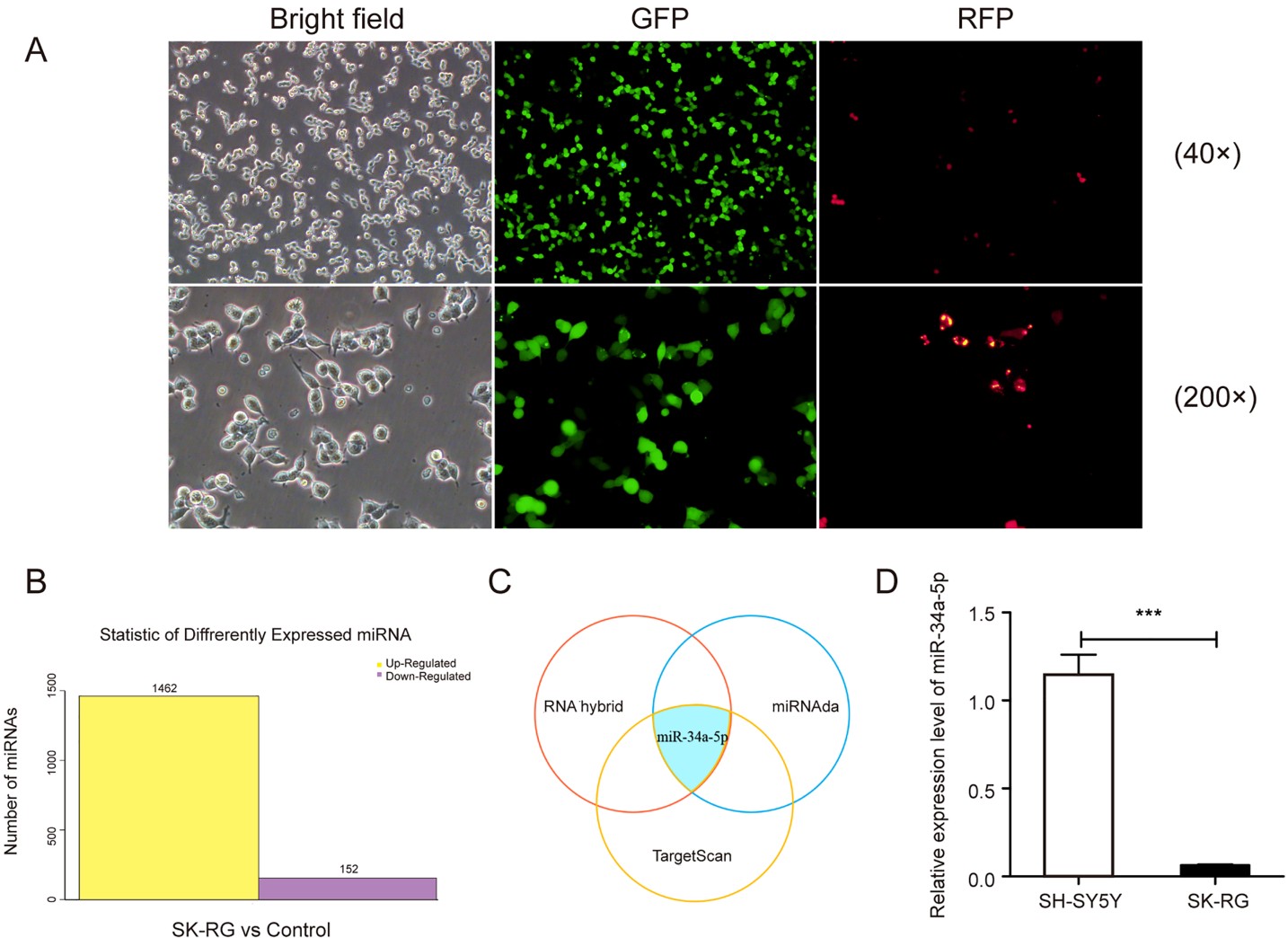

**Figure 1** **The miR-34a-5p levels were reduced in SK-RG cells.** (A) SH-SY5Y cells were infected with KSHV and re-named as SK-RG cells. Green fluorescent protein (GFP) fluorescence was used to confirm the infection (magnification, 40× and 200×). (B) The number of differentially expressed miRNAs in SK-RG cells. (C) Bioinformatics analysis to screen the differentially expressed miRNA intersection for regulating the c-fos. (D) The miR-34a-5p levels in SH-SY5Y and SK-RG cells were analyzed by real-time PCR. Data are presented as mean ± SD for three independent experiments. ***$P < 0.001$.

upregulated, and 152 were downregulated in SK-RG cells (Fig. 1B). We screened out some miRNA names with significant up-regulation and down-regulation, as shown in Table 1. The relationship between miRNA and mRNA expression profiles was analyzed using the TargetScan/miRanda/RNAhybrid software. miR-34a-5p expression was downregulated in SK-RG cells due to the intersection (Fig. 1C). Furthermore, we performed real-time PCR to confirm that miR-34a-5p was down-regulated in SK-RG cells (Fig. 1D).

## miR-34a-5p overexpression suppressed the proliferation and cell cycle of SK-RG cells

miR-34a-5p mimics were transfected into SK-RG cells to evaluate the effects of miR-34a-5p overexpression on cell proliferation and the cell cycle. MTT assay results

showed that miR-34a-5p overexpression inhibited the proliferation of SK-RG cells compared with that in the miR-NC group ($P < 0.05$, Fig. 2A). To investigate whether miR-34a-5p affected the SK-RG cell cycle, we detected the cell cycle by FCM. The percentage of cells in the G0/G1 phase increased significantly after overexpression of miR-34a-5p compared with miR-NC. The results indicated that miR-34a-5p arrested the cell cycle of SK-RG cells in G0/G1 phase ($P < 0.05$, Figs. 2B and 2C). The expression of related cyclins was evaluated by Western blot analysis. The levels of CDK4, CDK6, and cyclin D1 were decreased when miR-34a-5p was overexpressed ($P < 0.05$, Figs. 2D–2F). Using the plate clone assay, we also found that miR-34a-5p overexpression reduced the number of colonies in SK-RG cells ($P < 0.05$, Fig. 2G).

The miR-34a-5p inhibitor was transfected into SH-SY5Y cells to investigate the effect of miR-34a-5p knock-down on the proliferation of SH-SY5Y cells. The real-time PCR results indicated that the miR-34a-5p inhibitor successfully decreased its level ($P < 0.05$, Fig. S1A). MTT assay results showed increased cell proliferation when miR-34a-5p was knock-downed in SH-SY5Y cells ($P < 0.05$, Fig. S1B). These results suggested that miR-34a-5p can influence cell proliferation ability.

## miR-34a-5p overexpression inhibited the migration of SK-RG cells

We next explored whether miR-34a-5p mimics impacted the cell migration of SK-RG cells. The cell wound healing assay results showed the relative gap width was wider in miR-34a-5p mimics group than that in the miR-NC group ($P < 0.05$, Figs. 3A and 3B). In the transwell assay, the transmembrane cell number was fewer in miR-34a-5p mimics group than that in the miR-NC group ($P < 0.05$, Figs. 3C and 3D). Western blot results also revealed miR-34a-5p overexpression reduced the expression of MMP2 and MMP9 ($P < 0.05$, Figs. 3E and 3F). These data indicated that miR-34a-5p overexpression inhibited cell migration.

The miR-34a-5p inhibitor was transfected into SH-SY5Y cells to investigate the effect of miR-34a-5p knock-down on the migration of SH-SY5Y cells. Compared with the inhibitor-NC group, the miR-34a-5p inhibitor group accelerated the migration abilities of SH-SY5Y cells in wound-healing and transwell assays ($P < 0.05$, Figs. S1C–S1E). These results showed that miR-34a-5p could influence cell migration ability.

## miR-34a-5p regulated the expression of c-fos through the 3′-UTR

We employed the StarBase 2.0 database to predict the potential binding sequence of miR-34a-5p and c-fos 3′-UTR (Fig. 4A). In the dual-luciferase reporter gene assay, the co-transfection of miR-34a-5p mimics and the c-fos 3′-UTR-wt decreased luciferase activity. However, the co-transfection of the c-fos 3′-UTR-mut and miR-34a-5p mimics did not change any activity ($P < 0.05$, Fig. 4B). We compared the expression levels of c-fos in KSHV-infected and uninfected SH-SY5Y cells by Western blot assay. The results showed that c-fos was upregulated in SK-RG cells ($P < 0.05$, Fig. 4C). Furthermore, the real-time PCR displayed that the c-fos mRNA level was negatively correlated with the level of miR-34a-5p in SK-RG cells ($P < 0.05$, r = −0.9990, Fig. 4D). Besides, Western blot analysis showed that c-fos protein expression was down-regulated after miR-34a-5p

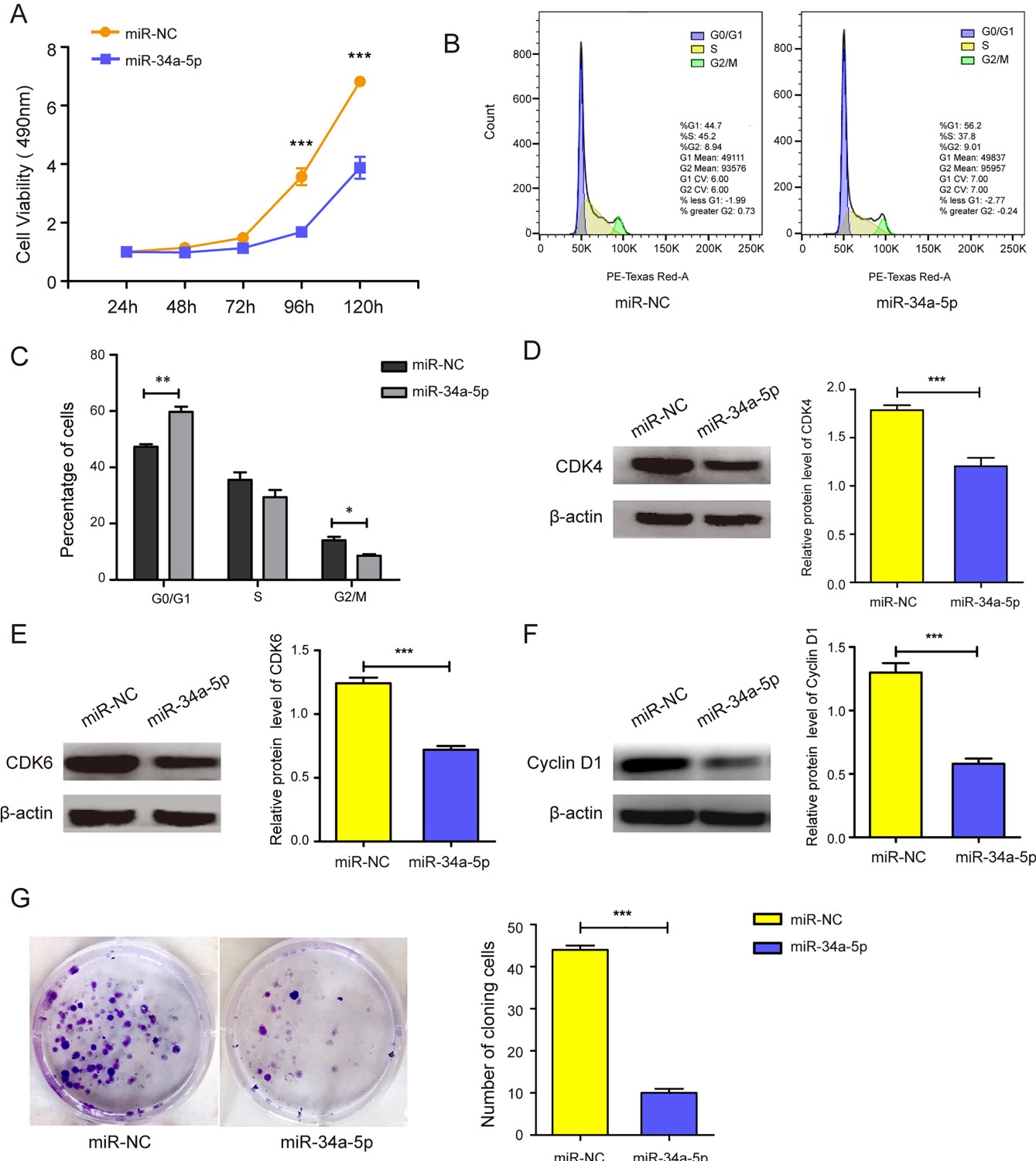

**Figure 2 miR-34a-5p inhibited the cell proliferation and the cell cycle in SK-RG cells.** (A) The effect of miR-34a-5p overexpression on proliferation was detected by the MTT assay. (B and C) The cell cycle was analyzed by the FCM in SK-RG cells after miR-34a-5p overexpression. The proportions of cells in the G0/G1, G2, and S phases were shown. (D–F) The protein expression levels of CDK4, CDK6, and cyclin D1 were detected by Western blot. (G) The effect of miR-34a-5p overexpression on colony formation was detected by plate clone assay. Data are presented as mean ± SD for three independent experiments. $^{*}P < 0.05$; $^{**}P < 0.01$; $^{***}P < 0.001$.

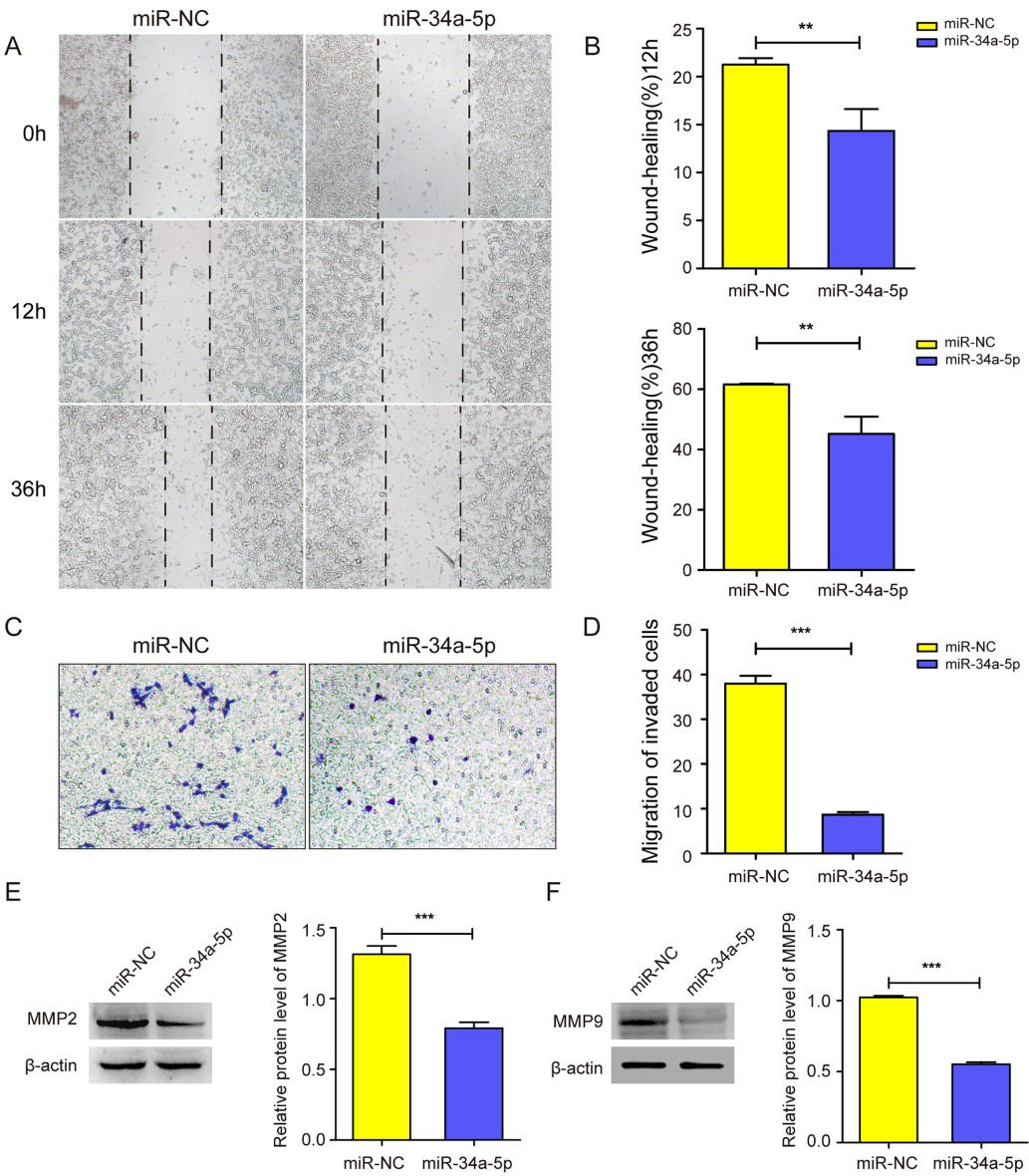

**Figure 3 miR-34a-5p inhibited the cell migration in SK-RG cells.** (A and B) The effect of miR-34a-5p overexpression on migration was detected by the wound healing assay. (C and D) The effect of miR-34a-5p overexpression on the migration was detected by the transwell assay. (E and F) Protein levels of MMP2 and MMP9 expression were detected by Western blot. Data are presented as Means ± SD for three independent experiments. **$P < 0.01$; ***$P < 0.001$.

overexpression ($P < 0.05$, Fig. 4E). These data indicated that miR-34a-5p had a targeting effect on the expression of c-fos.

## c-fos rescued the suppressive effect of miR-34a-5p on the proliferation of KSHV-infected SH-SY5Y cells

c-fos was upregulated in miR-34a-5p-overexpressing SK-RG cells ($P < 0.05$, Fig. 5A). Then the effect on the cell cycle was detected by FCM. After overexpression of c-fos, the percentage of cells in the G0/G1 phase was reduced compared with the control group,

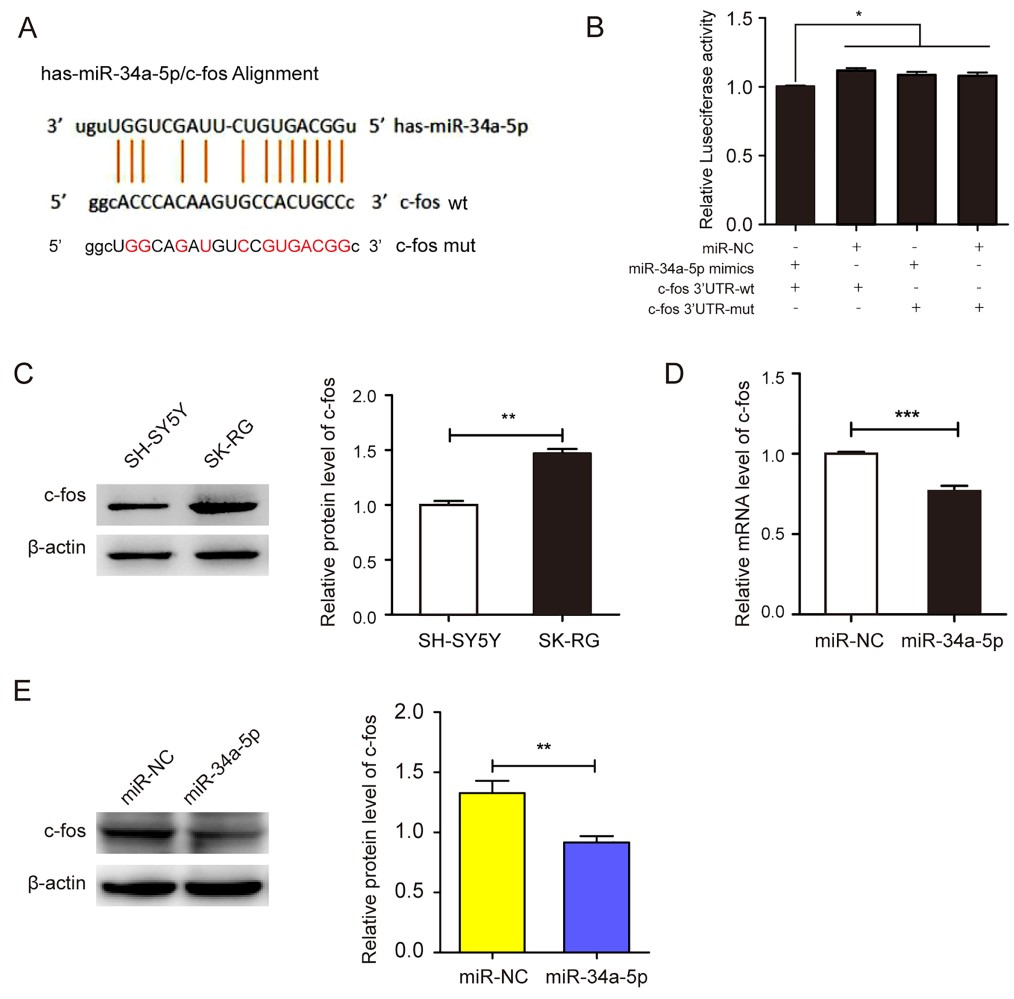

**Figure 4 miR-34a-5p had a targeting effect on the c-fos 3′-UTR.** (A) Potential binding sequences of miR-34a-5p and the c-fos 3′-UTR were predicted using the StarBase 2.0 database. The mutated were also shown. (B) The targeting relationship between miR-34a-5p and c-fos was confirmed by the dual-luciferase reporter gene assay. (C) The expression level of c-fos in KSHV-infected and uninfected SH-SY5Y cells by Western blot. (D) The mRNA expression level of c-fos was analyzed by real-time PCR assay in miR-34a-5p upregulated SK-RG cells. (E) The protein expression level of c-fos was identified in miR-34a-5p upregulated SK-RG cells by Western blot. Data are presented as Mean ± SD for three independent experiments. *$P < 0.05$; **$P < 0.01$; ***$P < 0.001$.

which indicated that c-fos reversed the cycle arresting effect of miR-34a-5p on SK-RG cells ($P < 0.05$, Figs. 5B and 5C). Subsequently, we explored the c-fos rescue effect on cell proliferation. MTT assays showed that c-fos overexpression rescued and reduced the inhibitory effect of miR-34a-5p on cell proliferation ($P < 0.05$, Fig. 5D). The protein levels of CDK4, CDK6, and cyclinD1 were increased when c-fos was overexpressed ($P < 0.05$, Figs. 5E–5G). Using the plate clone assay, we found that overexpression c-fos increased the colony number of SK-RG cells ($P < 0.05$, Figs. 5H and 5I). These results suggested c-fos rescued the effect of miR-34a-5p on the proliferation of SK-RG cells.

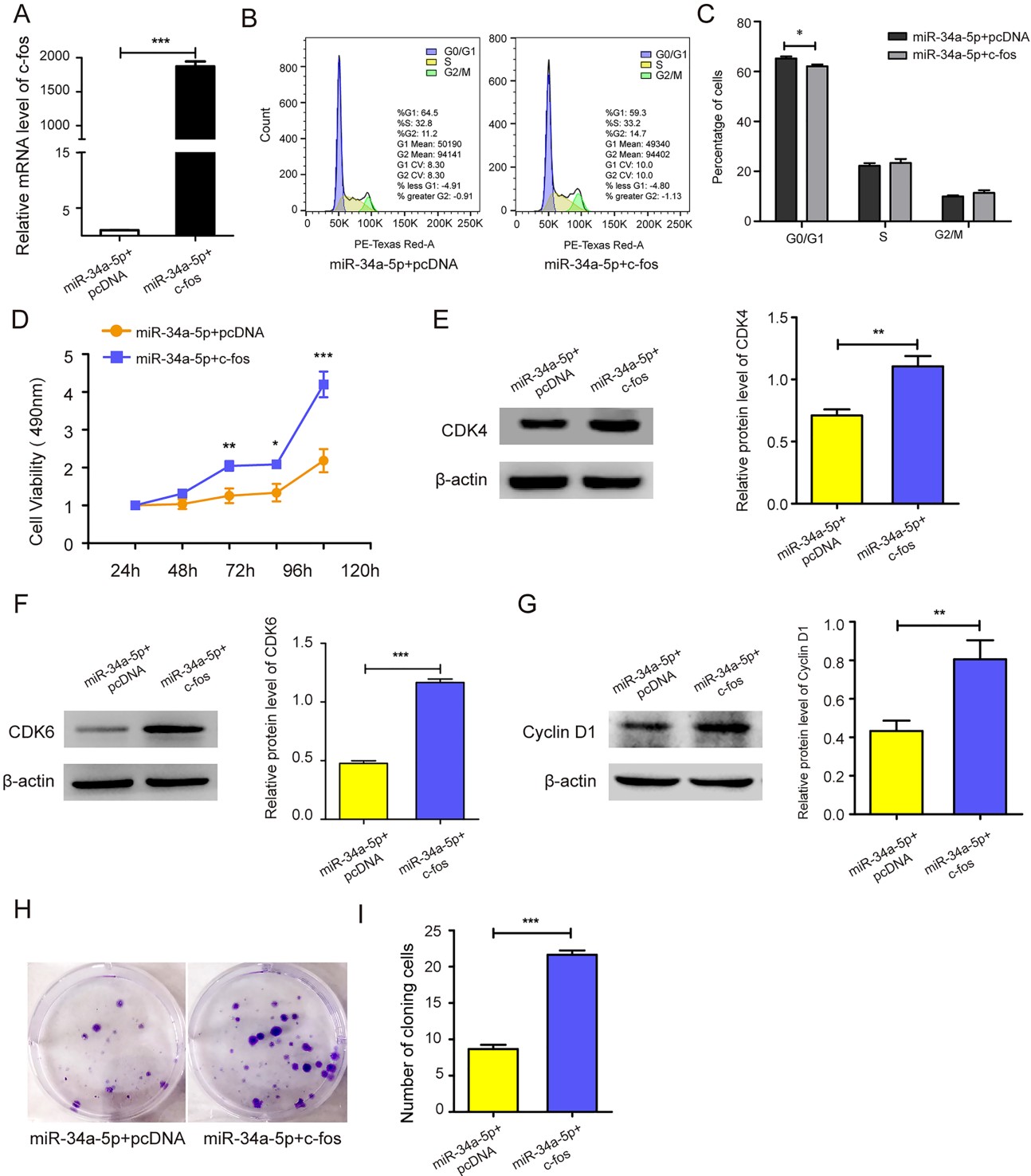

**Figure 5** **c-fos rescued the effect of miR-34a-5p on the proliferation of SK-RG cells.** (A) The effect of mRNA level of c-fos was analyzed by real-time PCR in miR-34a-5p upregulated SK-RG cells, which were transfected with the c-fos overexpression vector. (B and C) Cell cycle was examined by the FCM in SK-RG cells after overexpression of miR-34a-5p and c-fos. The proportions of cells in the G0/G1, G2, and S phases were shown. (D) The effect of c-fos on proliferation in the rescue assay was investigated by the MTT assay. (E-G) Protein levels of CDK4, CDK6, and cyclin D1 were detected using Western blot analysis in SK-RG cells. (H and I) Results of the plate clone assay in rescue assay. Data are presented as Means ± SD for three independent experiments. $^{*}P < 0.05$; $^{**}P < 0.01$; $^{***}P < 0.001$.

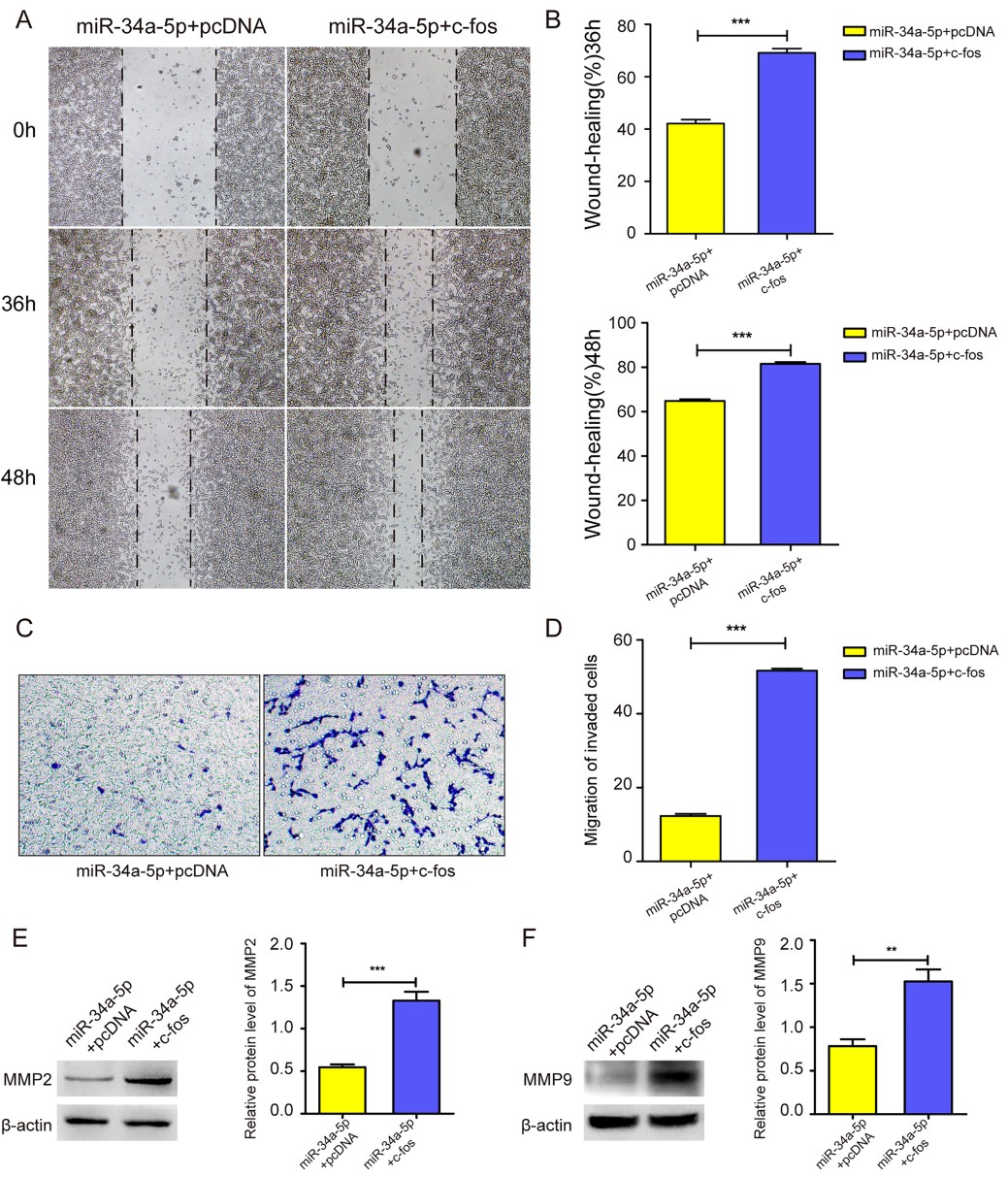

**Figure 6 c-fos rescued the effect of miR-34a-5p on the migration of SK-RG cells.** (A–D) The effect of c-fos on the migration in the rescue assay was detected by the wound healing assay (A and B) and transwell assay (C and D). (E and F) Protein expression levels of MMP2 and MMP9 were detected by Western blot. Data are presented as Mean ± SD for three independent experiments. **$P < 0.01$; ***$P < 0.001$.           

## c-fos rescued the suppressive effect of miR-34a-5p on the migration of KSHV-infected SH-SY5Y cells

We upregulated c-fos in miR-34a-5p-overexpressing SK-RG cells, and the cell migration ability was detected. The cell wound healing assay showed that the gap was narrower in the c-fos rescue group than in the control group ($P < 0.05$, Figs. 6A and 6B). Similarly, the transwell assay showed that the overexpression of c-fos rescued this effect, although miR-34a-5p suppressed the migration of SK-RG cells. ($P < 0.05$, Figs. 6C and 6D). Western

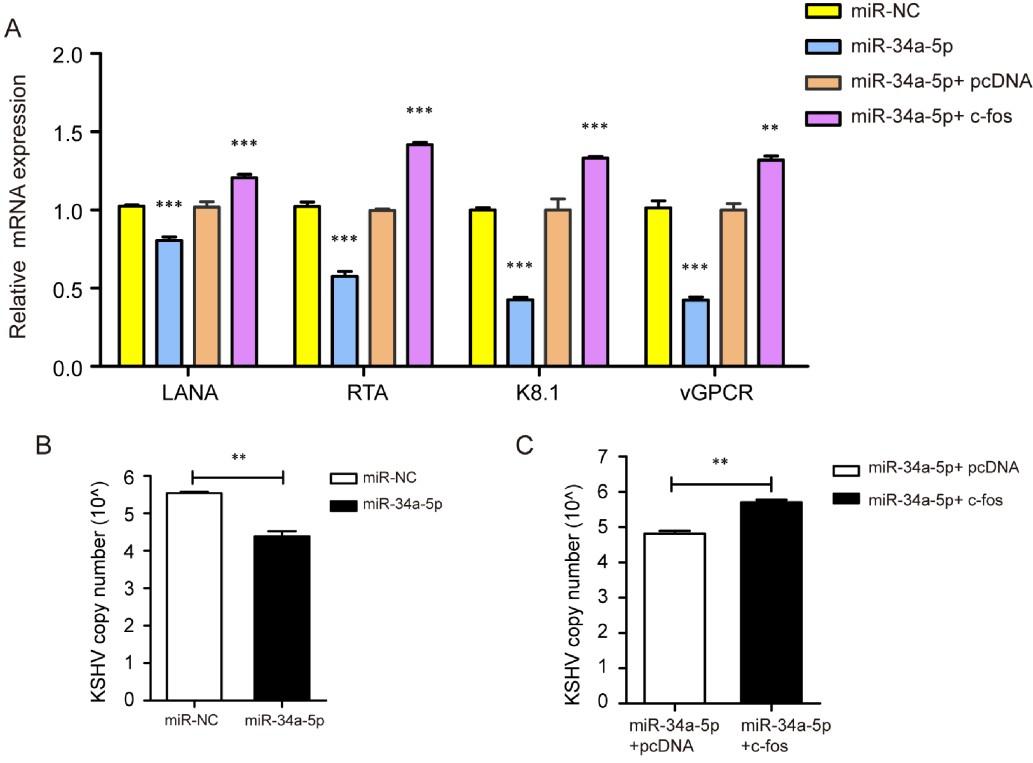

**Figure 7** **miR-34a-5p decreased the KSHV copy number and virus gene expression, and c-fos rescued the suppressive role.** (A) The mRNA expression levels of LANA, RTA, K8.1, and v-GPCR were analyzed by real-time PCR in the miR-34a-5p upregulated and rescue groups. (B) KSHV virus DNA copy number was detected by the Taq-man real-time PCR after overexpression of miR-34a-5p in SK-RG cells. (C) KSHV DNA copy number in the rescue assay was detected by the Taq-man real-time PCR. Data are presented as mean ± SD for three independent experiments. $^{**}P < 0.01$; $^{***}P < 0.001$.

blot analysis showed that the protein levels of MMP2 and MMP9 increased when c-fos was overexpressed ($P < 0.05$, Figs. 6E and 6F). In summary, c-fos rescued the effect of miR-34a-5p on the migration of SK-RG cells.

## miR-34a-5p decreased the KSHV gene expression and copy number, while c-fos rescued the suppressive effects

We transfected miR-NC and miR-34a-5p to investigate the effect of miR-34a-5p on KSHV gene expression and copy number. The results indicated that miR-34a-5p overexpression suppressed the mRNA expression of KSHV genes LANA, RTA, K8.1, and v-GPCR. Furthermore, we upregulated c-fos in miR-34a-5p-overexpressing SK-RG cells and found that the overexpression of c-fos reversed the effects of miR-34a-5p on these KSHV genes' expressions ($P < 0.05$, Fig. 7A). We determined the KSHV DNA copy number after overexpression of miR-34a-5p by Taq-man real-time PCR. The results showed miR-34a-5p overexpression decreased the KSHV copy number ($P < 0.05$, Fig. 7B). However, after we upregulated the c-fos, the c-fos rescued the restraint effect of miR-34a-5p on KSHV copy number ($P < 0.05$, Fig. 7C).

### miR-34a-5p inhibited tumor proliferation *in vivo*

Xenograft tumor mouse models were established by injecting SK-RG cells into one side of the spine of nude mice to investigate the efficacy of miR-34a-5p *in vivo* (Fig. 8A). The mice were treated with miR-34a-5p agomir and miR-NC agomir. miR-34a-5p agomir treatment led to a suppression of the growth of tumors, compared with that in the miR-NC group (Figs. 8B and 6C). This consistent trend was also demonstrated by the visual inspection of the isolated tumor size and quantitative measurement of the tumor weight and volume. The tumor volume decreased in mice treated with miR-34a-5p compared with that in the miR-NC group ($P < 0.05$, Fig. 8D). Moreover, we used immunohistochemistry (IHC) to evaluate the expression levels of c-fos, CDK6, and cyclin D1 in the tumor tissues. The results showed that the levels of c-fos, cyclin D1, and CDK6 were lower in the tissue sections treated with miR-34a-5p than those in the miR-NC group ($P < 0.05$, Figs. 8E–8H). Tumor weight decreased in mice treated with miR-34a-5p compared with that in the miR-NC group ($P < 0.05$, Fig. 8I). The data suggested that miR-34a-5p inhibited proliferation *in vivo*.

## DISCUSSION

Increasing studies have shown that miRNAs encoded by the host or virus and their target genes together form a new regulatory network between the host and the virus. miRNAs are involved in cancer development and serve as molecular biomarkers for cancer diagnosis, treatment, and prognosis. Some of the downregulated miRNAs function as tumor suppressor genes, while upregulated miRNAs have pro-tumor activity (*Petrek et al., 2019*; *Vishnoi & Rani, 2017*; *Yu et al., 2019*).

By screening for differentially expressed miRNA, we found that the miR-34a-5p level was down-regulated in KSHV-infected neuronal cells, and miR-34a-5p might regulate the important transcription factor c-fos. Hence, miR-34a-5p was a candidate miRNA worthy of further exploration. In human papillomavirus (HPV)-infected human epidermal keratinocytes, miR-34a-5p inhibited cell proliferation and migration by targeting the Jagged 1 (JAG1) Notch1 pathway (*Gao et al., 2020*). Also, miR-34a-5p inhibited tumorigenesis and progression of gliomas by targeting high-mobility group AT-hook 2 (HMGA2) (*Ma et al., 2019*). However, the pathogenesis of miR-34a-5p in KSHV is unclear.

We analyzed the effects of miR-34a-5p on the proliferation and migration of SK-RG cells by upregulating the miR-34a-5p level. The results showed that miR-34a-5p inhibited the proliferation and migration of SK-RG cells.

The function of miR-34a-5p may be achieved by regulating the target gene. Bioinformatics prediction analysis showed that c-fos might be the target gene of miR-34a-5p. Furthermore, c-fos is a marker of neuronal activation and plays an important role in neuronal activation, synaptic plasticity, and neuronal apoptosis (*Zeng et al., 2007*). It has also been shown to regulate the expression of KSHV genes. When subjected to extracellular stimulation, c-fos can be rapidly induced to cause transcriptional expression and form a heterodimer (transcription factor activator protein 1, AP-1) with c-Jun protein (*Alfonso-Gonzalez & Riesgo-Escovar, 2018*; *Tsiambas et al., 2020*). This heterodimer can

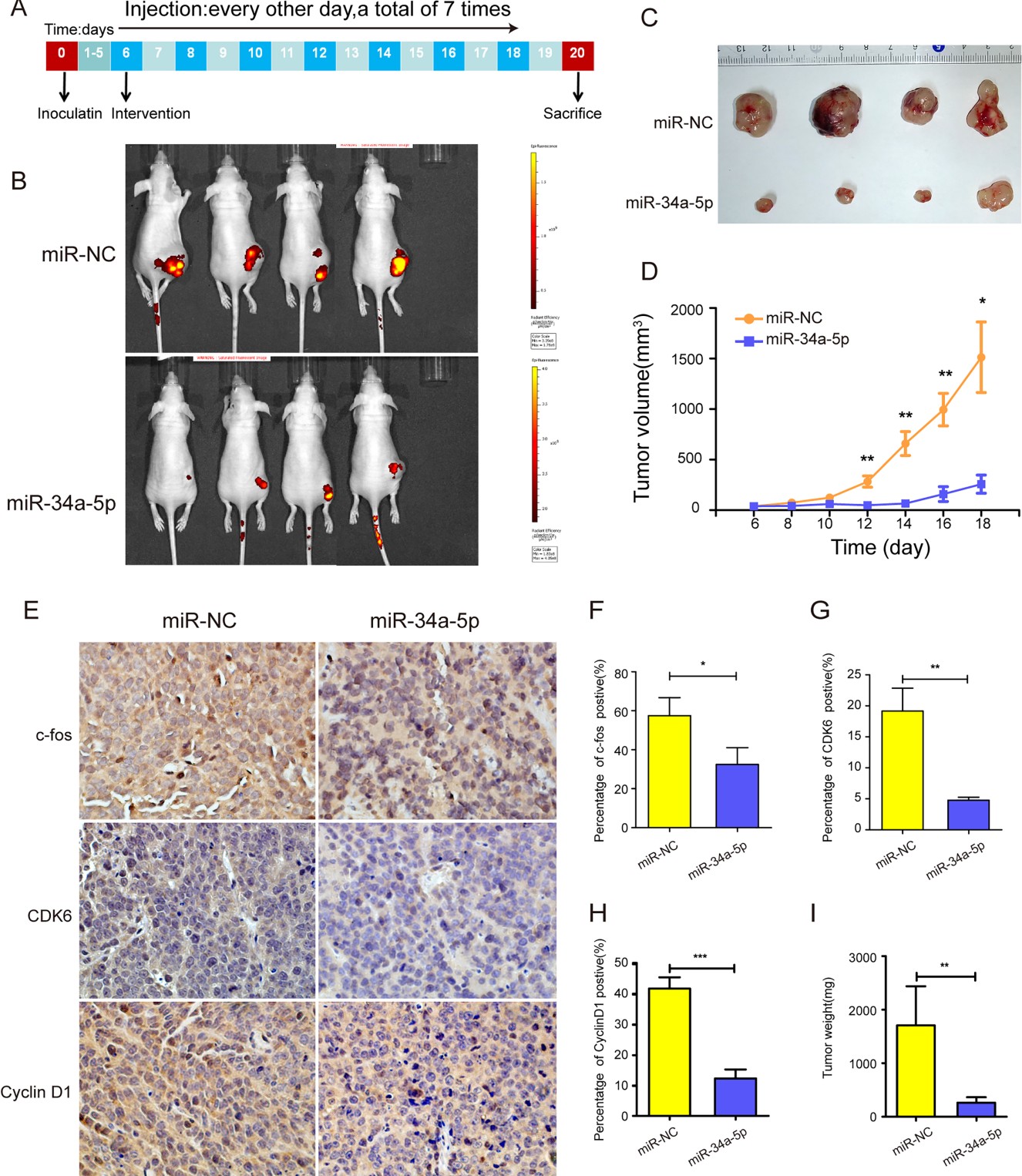

**Figure 8 miR-34a-5p effectively inhibited tumor proliferation in nude mice xenograft tumor models.** (A) Protocol for establishing *in vivo* studies of xenograft tumor models. (B) The invivoimagingsystem (IVIS) LUMina III system was used to detect tumors in nude mice. (C) Visual inspection of the isolated tumor size. (D) The volume (mm³) of the tumor was measured using vernier calipers in the miR-34a-5p agomir and miR-NC agomir treatment groups (miR-NC: 364.95 ± 202.52; miR-34a-5p: 2079.03 ± 536.69). (E) Levels of c-fos, CDK6, and cyclin D1 in tumor

**Figure 8 (continued)**
tissue samples of nude mice were detected by IHC analyses. (F–H) Quantification of the relative level of c-fos, CDK6, and cyclin D1. (I) Comparison of the weight (mg) of the xenograft tumors (miR-NC: 1708 ± 631.74; miR-34a-5p: 262.25 ± 88.52). Data are presented as means ± SD for three independent experiments. $^*P < 0.05$; $^{**}P < 0.01$; $^{***}P < 0.001$. 

bind to the DNA sequence 5′-TGAGCTCA-3′ of the 12-O-tetradecanoyl-phorbol-13-acetate (TPA) response element. TPA is an important inducer of the lytic state of KSHV *in vitro* (*Cohen, Brodie & Sarid, 2006*; *Santarelli et al., 2019*). The expression of c-fos causes the TPA-like cleavage of KSHV, which may be one of the underlying mechanisms associated with c-fos-induced KSHV lysis. In KSHV-infected epithelial cells, c-fos was reported to promote the transcription of viral genes through direct binding to multiple KSHV promoters, which induced the sustained activation of extracellular signal-regulated kinase (ERK)-mitogen-activated protein kinase (MAPK), a signaling pathway closely related to cell proliferation (*Milde-Langosch, 2005*).

Our previous study also found that the expression of c-fos was upregulated in KSHV-infected neuronal cells and affected the pathogenicity of the virus and cell proliferation (*Xu, Cao & Li, 2020*). Therefore, the effects of c-fos on the proliferation of KSHV-infected SH-SY5Y need further investigation. miRNAs exhibit their biological functions by regulating the targets through 3′-UTR interactions (*Pasquinelli, 2012*). *Deng et al. (2021)* reported that miR-34a-5p reversed multidrug resistance in gastric cancer cells by targeting the 3′-UTR of Sirtuin 1 (SIRT1) and inhibiting its expression. *Wang et al. (2020)* demonstrated that miR-34a-5p is bound to the 3′-UTR of lymphoid enhancer-binding factor 1 (LEF1). miR-34a-5p overexpression inhibited the proliferation and invasion of esophageal squamous cell carcinoma (ESCC) cells. The rescue experiments showed that the re-expression of LEF1 reversed the inhibitory effect of miR-34a-5p (*Wang et al., 2020*). Similarly, we predicted the binding site between miR-34a-5p and the c-fos 3′-UTR using bioinformatics StarBase 2.0 software. The dual-luciferase reporter gene assay and c-fos expression detection in miR-34a-5p-overexpressing cells further confirmed the targeted regulation of c-fos by miR-34a-5p.

We transfected c-fos plasmid into miR-34a-5p-overexpressing cells. In the rescue assay, we found that the overexpression of c-fos reversed the inhibitory effect of miR-34a-5p. At the same time, we constructed the xenograft tumor model. *In vivo* experiments further proved that miR-34a-5p effectively inhibited tumor growth. IHC of tumor tissues showed that the expression of c-fos and cell cycle proteins was downregulated under the miR-34a-5p treatment.

In conclusion, we proved that the miR-34a-5p level was down-regulated in SK-RG cells. The upregulated miR-34-5p inhibited malignant behaviors virus gene expression and down-regulated the KSHV copy number targeting c-fos in KSHV infected SH-SY5Y cells. Hence, miR-34a-5p is the candidate molecular drug for controlling the malignant progression of KSHV-infected neuronal cells. The upstream regulatory mechanisms of miR-34a-5p need further exploration.

### Funding

This study was supported by the Fund of National Natural Science Foundation of China (NSFC81760362), the International Cooperation Program of Shihezi University (GJHZ202102), the Xinjiang Autonomous Region Postgraduate Research and Innovation Project (XJ2020G124, XJ2021G122) and the National College Students Innovation and Entrepreneurship Training Program (202110759020). The funders had a role in study design, data collection and analysis. The funders had no role in the decision to publish or the preparation of the manuscript.

### Grant Disclosures

The following grant information was disclosed by the authors:
National Natural Science Foundation of China: NSFC81760362.
International Cooperation Program of Shihezi University: GJHZ202102.
Xinjiang Autonomous Region Postgraduate Research and Innovation Project: XJ2020G124, XJ2021G122.
National College Students Innovation and Entrepreneurship Training Program: 202110759020.

### Competing Interests

The authors declare that they have no competing interests.

### Author Contributions

- Shuyuan Wu performed the experiments, analyzed the data, prepared figures and/or tables, and approved the final draft.
- Zhaofu Wu performed the experiments, prepared figures and/or tables, and approved the final draft.
- Huiling Xu performed the experiments, prepared figures and/or tables, and approved the final draft.
- Jinli Zhang analyzed the data, prepared figures and/or tables, and approved the final draft.
- Wenyi Gu conceived and designed the experiments, authored or reviewed drafts of the paper, and approved the final draft.
- Xiaohua Tan analyzed the data, authored or reviewed drafts of the paper, and approved the final draft.
- Zemin Pan conceived and designed the experiments, authored or reviewed drafts of the paper, and approved the final draft.
- Dongdong Cao conceived and designed the experiments, performed the experiments, prepared figures and/or tables, and approved the final draft.
- Dongmei Li performed the experiments, prepared figures and/or tables, and approved the final draft.

- Lei Yang analyzed the data, authored or reviewed drafts of the paper, and approved the final draft.
- Dongmei Li conceived and designed the experiments, authored or reviewed drafts of the paper, and approved the final draft.
- Yuanming Pan conceived and designed the experiments, authored or reviewed drafts of the paper, and approved the final draft.

## Animal Ethics

The following information was supplied relating to ethical approvals (*i.e.*, approving body and any reference numbers):

Approval Letter of Animal Experimental Ethical Inspection of First Affiliated Hospital, Shihezi University School of Medicine.

## Data Availability

The raw measurements are available in the Figures and the Supplemental File.

## Supplemental Information

Supplemental information for this article can be found online at http://dx.doi.org/10.7717/peerj.13233#supplemental-information.

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
