# Peer review of "miR-34a-5p inhibits the malignant progression of KSHV-infected SH-SY5Y cells by targeting c-fos"

_PeerJ, doi:10.7717/peerj.13233_

## Round 0.1 · original submission · Major Revisions

Which miRNAs are most upregulated and downregulated in the volcano plot, Fig 1?

When studying effects on cell cycle, a cell cycle analysis should be performed.

Correct statistical tests should be used for the analysis of Fig. 4b.

The manuscript should be corrected by a proficient speaker.

·

Basic reporting

The manuscript entitled “Study of miR-34a-5p inhibiting the malignant progression of KSHV-infected SH-SY5Y cells by targeting c-fos” by Shuyuan Wu et al. investigated the effects of miR-34a-5p on growth and motility of SH-SY5Y cells infected with Kaposi's sarcoma−associated herpesvirus (KSHV) by regulating c-fos.

Experimental design

1. The study lacks a control of miR-34a-5p-depleted SH-SY5Y cells. Can miR-34a-5p knockdown transform uninfected SH-SY5Y cells and increase cell growth and migration?
2. What is the sequence of c-fos 3-UTR-mut in Figure 4A?
3. The statistical method used in Figure 4B is improper. One-way ANOVA should be used to test whether it is statistically different among all compared groups. Multiple comparison is followed to test which two groups are different.
4. Figure 4C did not show the correlation between c-fos mRNA level and miR-34a-5p, as stated. To prove correlation, authors need to calculate correlation coefficient.
5. In Figure 7, mRNA expression data of each gene should be presented in the same panel. Therefore, panel A and B, panel C and D, panel E and F, and panel G and H should be in the same panel. It is expected that overexpression of miR-34a-5p inhibits KSHV genes.

Validity of the findings

Authors well characterized the phenotypes of SK-RG cells. Major phenotypes that were affected by miR-34a-5p should be reproduced and validated in a second neuronal cell line.

Additional comments

The manuscript will be improved by moderate English editing.

Reviewer 2 ·

Basic reporting

In this study, Wu et al aim to investigate the role of miR-34a-5p on the proliferation and migration of KSHV-infected SH-SY5Y cells and study how miR-34a-5p regulates c-fos. They found miR-34a-5p was decreased in SK-RG (KSHV-infected SH-SY5Y) cells. Increasing miR-34a-5p in SK-RG cells reduced its proliferation, migration, which could be associated with decrease of CDKs and MMPs. In addition, overexpression of c-fos (one target of miR-34a-5p) restored miR-34a-5p mimic-mediated decrease of proliferation, migration, CDKs, MMPs, and KSHV gene expression. In vivo, miR-34a-5p agomir reduced tumor formation. Generally speaking, this study is well-organized and well-designed. Following are the comments:
1. Language editing is needed. The grammatical errors significantly decreased the accuracy of the information that the authors wanted to deliver.
2. I have concern about the efficiency of rKSHV.219 infection. From Figure 1A, the successfully infected cells should carry GFP. However, from their data, the infection efficiency is not 100%. In this case, sorting the GFP positive cells is needed. Otherwise, wild type SH-SY5Y cells will be mixed into the SK-RG cells. Thus, the results will be less reliable.
3. The authors should at least label out the most upregulated and downregulated miRNAs in their volcano plot. Otherwise, why bother to show it?
4. I see under subhead 3.2 the authors wanted to study cell cycle. Then, cell cycle analysis would be recommended. Evidence directly shows that increase of miR-34a-5p leads to cell cycle arrest is needed.
5. Another issue of this study is the protein changes in western blot are not significant enough to convince people that those proteins were truly increased or decreased. This happened in Figure 3E-F, Figure 5F, and Figure 6E.
6. Did the authors run a power analysis to confirm the number of mice they needed for each treatment for their xenograft model?
7. Have the authors compared the c-fos expression in their SH-SY5Y and SK-RG cells? This is important.
8. It is unknow the regulatory pattern of miR-34a-5p, c-fos, and KSHV. Does miR-34a-5p regulates KSHV directly or the regulation is via c-fos? Further data are needed to clarify this.

Experimental design

Additional experiments are needed.

Validity of the findings

Most of the findings are sound.

---

## Round 0.2 · Minor Revisions

I agree with the second reviewer that the cell cycle analysis would be more convincing if the automatically generated frequency from the software to their FACS plot would be added.

·

Basic reporting

Authors have successfully responded to my comments.

Experimental design

None.

Validity of the findings

None.

Additional comments

None.

Reviewer 2 ·

Basic reporting

The authors improved their manuscript according to my comments. Now, there is just one issue remains. They did a cell cycle analysis according to my suggestion, which is good. However, from the FACS histogram, the change between two compared groups is marginal. Usually, FACS analysis software is capable of automatically labeling the frequency for each cell cycle phase. I would suggest the authors add the automatically generated frequency to their FACS plot. This will only make their data more convincing.

Experimental design

No comment

Validity of the findings

No comment

---

## Round 0.3 · accepted · Accept

Thanks for the final corrections, they are very helpful for the readership and make your data stronger.